# Elevated ozone disrupts mating boundaries in drosophilid flies

Nan-Ji Jiang ®[1,2], Xinqi Dong[1], Daniel Veit[3], Bill S. Hansson ®[1,2,4] & Markus Knaden ®[1,2,4] ✉

Animals employ different strategies to establish mating boundaries between closely related species, with sex pheromones often playing a crucial role in identifying conspecific mates. Many of these pheromones have carbon-carbon double bonds, making them vulnerable to oxidation by certain atmospheric oxidant pollutants, including ozone. Here, we investigate whether increased ozone compromises species boundaries in drosophilid flies. We show that short-term exposure to increased levels of ozone degrades pheromones of *Drosophila melanogaster*, *D. simulans*, *D. mauritiana*, as well as *D. sechellia*, and induces hybridization between some of these species. As many of the resulting hybrids are sterile, this could result in local population declines. However, hybridization between *D. simulans* and *D. mauritiana* as well as *D. simulans* and *D. sechellia* results in fertile hybrids, of which some female hybrids are even more attractive to the males of the parental species. Our experimental findings indicate that ozone pollution could potentially induce breakdown of species boundaries in insects.

Reproductive isolation, including pre- and post-mating isolation, is regarded as one of the main drivers for speciation[1–6]. Animals employ various strategies to maintain pre-mating isolation from closely related species. One well-known mechanism for finding conspecific partners and distinguishing them from closely related species is the use of sex pheromones, e.g. the release of chemical substances to attract conspecific mates[7]. In drosophilid flies sex pheromones are sometimes shared among close relatives[8–10], such as the cosmopolitan species *D. melanogaster* and *D. simulans*, and their close relatives *D. mauritiana* (endemic to Mauritius) and *D. sechellia* (endemic to the Seychelles)[11,12]. All these species have different amounts of the same male pheromones such as 11-cis-Vaccenyl acetate (cVA), (*Z*)−7-Tricosene (7-T), and (*Z*)−7-Pentasene (7-P) and partly differ in their female pheromones such as (*Z*)−7,11-Heptacosadiene (7,11-HD) and (*Z*)−7,11-Nonacosadiene (7,11-ND) in *D. melanogaster* and *D. sechellia*, and 7-T in *D. simulans*, and *D. mauritiana*[9,13–19]. For these sympatric closely related species, presence or absence of a given compound as well as its relative amount compared to other compounds drive an individual's attractiveness to conspecific and repulsiveness to allospecific mates. Therefore, these compounds contribute to maintaining the mating boundaries between the four *Drosophila* species[19].

Like most other insect sex pheromones, these compounds share a common characteristic: they contain carbon-carbon double bonds that can easily become oxidized by e.g. oxidant pollutants like ozone or nitric oxides[20]. Human activities, especially the combustion of fossil energies, have led to a rise in global atmospheric concentrations of oxidant pollutants like ozone and nitric oxides. The yearly average of e.g. the atmospheric ozone concentration has increased from pre-industrial levels of 5–10 parts per billion (ppb) to nowadays ~20–45 ppb and is predicted to rise by another 23% by 2050[21–24]. Urban areas, however, particularly in regions such as Mexico, China, the USA, Nigeria, Brazil, and India, already now experience ozone concentrations exceeding 100 ppb[25–31]. These pollution levels of ozone not only pose risks to humans but also have significant impacts on insect

[1]Department of Evolutionary Neuroethology, Max Planck Institute for Chemical Ecology, Hans-Knöll-Straße 8, D-07745 Jena, Germany. [2]Next Generation Insect Chemical Ecology, Max Planck Centre, Max Planck Institute for Chemical Ecology, Hans-Knöll-Straße 8, D-07745 Jena, Germany. [3]Max Planck Institute for Chemical Ecology, Hans-Knöll Straße 8, D-07745 Jena, Germany. [4]These authors contributed equally: Bill S. Hansson, Markus Knaden. ✉e-mail: mknaden@ice.mpg.de

populations[32-34]. We recently could show that short-term exposure to 100 ppb levels of ozone is enough to degrade sex pheromones in many drosophilid species and can disrupt their sex communication[20]. Male flies that usually attract female conspecifics by emitting a species-specific blend of pheromones, became less attractive, as large parts of their pheromones became degraded by ozone[20]. However, these pheromone blends do not only govern attraction for conspecifics, but are also involved in the establishment of species boundaries by repulsing potential allospecific mates[13,19,35]. Despite these boundaries, interspecific gene flow between populations of some of the species has been observed[36,37]. Here we show that currently observed increased levels of ozone are sufficient to break down mating boundaries between closely related *Drosophila* species in laboratory conditions, and can lead to increased levels of hybridization. Many of the resulting hybrids are sterile and, hence, represent an evolutionary dead end. However, hybridization of some sympatric species combinations results in fertile hybrids, which are as or more attractive than their purebred counterparts in mating choice assays. This could potentially lead to a continued gene flow between two closely related species. Hence, our results suggest that, by degrading pheromones, anthropogenic oxidant pollutants may compromise mating boundaries in insects.

## Results and discussion
### Ozone-exposed flies carry lower amounts of pheromones
We exposed males and females of *D. melanogaster*, *D. simulans*, *D. mauritiana*, and *D. sechellia* for 2 h to 100 ppb of ozone, i.e. to conditions that have been frequently observed in urban areas[25-31], and subsequently analyzed body odors of single flies using a thermal desorption unit coupled with gas chromatography-mass spectrometry

(TDU GC-MS, Fig. 1a, b). The comparison with control flies, that were exposed for 2 h to ambient air (which in our case contained about 5 ppb ozone[20]), resulted in a significant decrease in the amounts of male pheromones, as well as a reduction in the levels of female compounds (Fig. 1b, c). Analyzing the recovery rates of pheromone levels after ozone exposure resulted in still significant differences after 1 day, that, however, disappeared after 2 days (Fig. S1). Hence, it takes at least 2 days for the pheromone levels to recover. As peaks of ozone concentrations often re-occur on a daily base[26], flies in polluted areas might not be able to re-establish their pheromone levels in-between.

### Ozone exposure can corrupt pheromone-dependent species boundaries
Although male *Drosophila* flies are known to frequently court allospecific females[38,39], the differences in sex pheromones between the different species of the *D. melanogaster* subgroup are known to constitute species boundaries[19]. We next asked, whether the mating barriers between the different species can be corrupted by the ozone-induced decrease of pheromones. Whether or not a male fly succeeds in mating depends on its acceptance by the female, which again is known to depend on the species-specific blend of its pheromones[19]. Considering that all compounds of the *D. melanogaster* subgroup that are known to be involved in this process (cVA, 7-T, 7-P, 7,11-HD, and 7,11-ND) can be oxidized by ozone (Fig. 1), we then conducted mate-choice assays with those species combinations that in nature occur sympatrically to determine whether hybridization events would increase in number after exposing the flies to ozone. As ozone exposure generally leads to lower acceptance of males by female flies[20], we designed a two-choice test, where the flies had a long time to court and mate. Males and females from two different species were first exposed

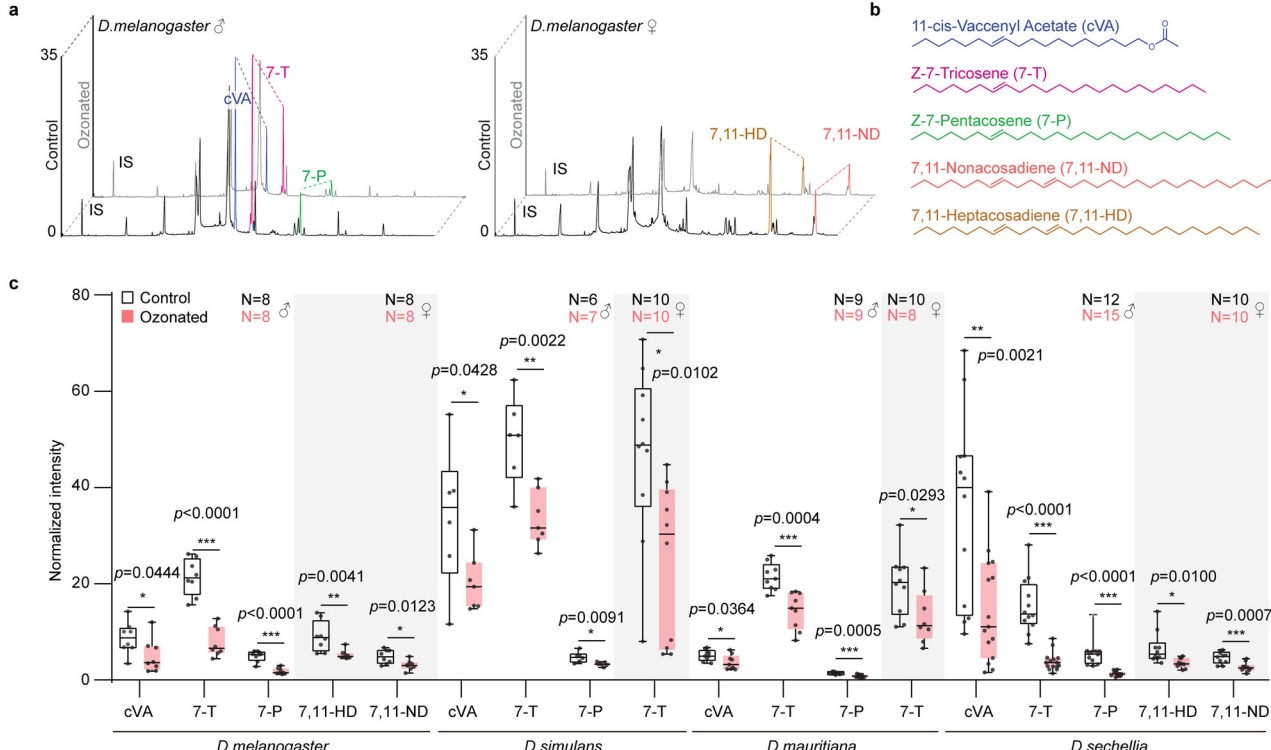

**Fig. 1 | TDU GC-MS analysis of pheromones and cuticular hydrocarbons of both sexes from *D. melanogaster* (CS), *D. simulans*, *D. mauritiana*, and *D. sechellia*.**
**a** Examples of chemical profiles of ozonated (grey) or control (black) male and female *D. melanogaster* flies. Ozonated (or control), 2 h exposure to 100 ppb ozone (or ambient air with about 5 ppb ozone) directly before TDU-GC analysis. IS: internal standard. **b** Chemical structures of described pheromones of the

*D. melanogaster* subgroup. **c** Quantitative analysis of the same compounds for four *Drosophila* species. The box plots present median values and quartiles, whiskers the minimum and maximum values, and dots the individual data points. White or pink plots indicate control or ozonated treatment, respectively. Two-side unpaired *t* test, *p < 0.05; **p < 0.01; ***p < 0.001.

separately for two hours to either a high level of 100 ppb (ozonated flies) or ambient air (control flies). Afterwards a single female was confronted for 6 h with one conspecific and one allospecific male (both exposed to the same levels of ozone as the female) and was afterwards allowed to lay its eggs. As females of these four species do not re-mate within 6 h after mating (*D. melanogaster*[40,41]; *D. simulans*[41], *D. sechellia*, and *D. mauritiana*, see Fig. S2), the identification of offspring informed us about the identity of the successful male (Fig. 2a). Previous studies have demonstrated that in the case of hybridization between *D. melanogaster* and the other three species *D. simulans*, *D. mauritiana*, or *D. sechellia*, the resulting hybrids exhibit a single sex, which depending on the species combination is either male or female[42,43]. In contrast, hybrids within the other three species display traditional binary sexes, but male hybrids can be distinguished from non-hybrids by examining the posterior lobe morphology of their genitalia[44–46]. To identify the offspring from the above experiments, we, hence, performed no-choice assays between all species that potentially can hybridize in nature, i.e. we excluded crossings of *D.*

*mauritiana* and *D. sechellia* that inhabit different islands and do not exhibit any sympatric population. From the pure species and the gained hybrid males, we constructed a posterior lobe *atlas* (Fig. S3). Contrary to ref. 44,42 but in line with ref. 47 we did not obtain any hybrids from no-choice assays using *D. mauritiana* females with *D. simulans* males. Nevertheless, we compared the genitalia of males obtained from choice assays between both species with the hybrid morphology reported by ref. 44 and to the purebred *D. mauritiana*.

The results from two-choice tests revealed that hybrids were extremely rare or even absent when flies were exposed to ambient air before the females of the different species could choose between a conspecific and an allospecific male. Only two of ten combinations (Fig. 2b, d) resulted in at least few hybridization events. However, when both sexes were exposed to 100 ppb of ozone before the mate-choice experiment, we observed hybridization in seven out of ten cases (Fig. 2b–e) and a significant increase in three of these combinations (Fig. 2b and d). Specifically, in the case where *D. simulans* females were confronted with a conspecific and a *D. mauritiana* male, ozone

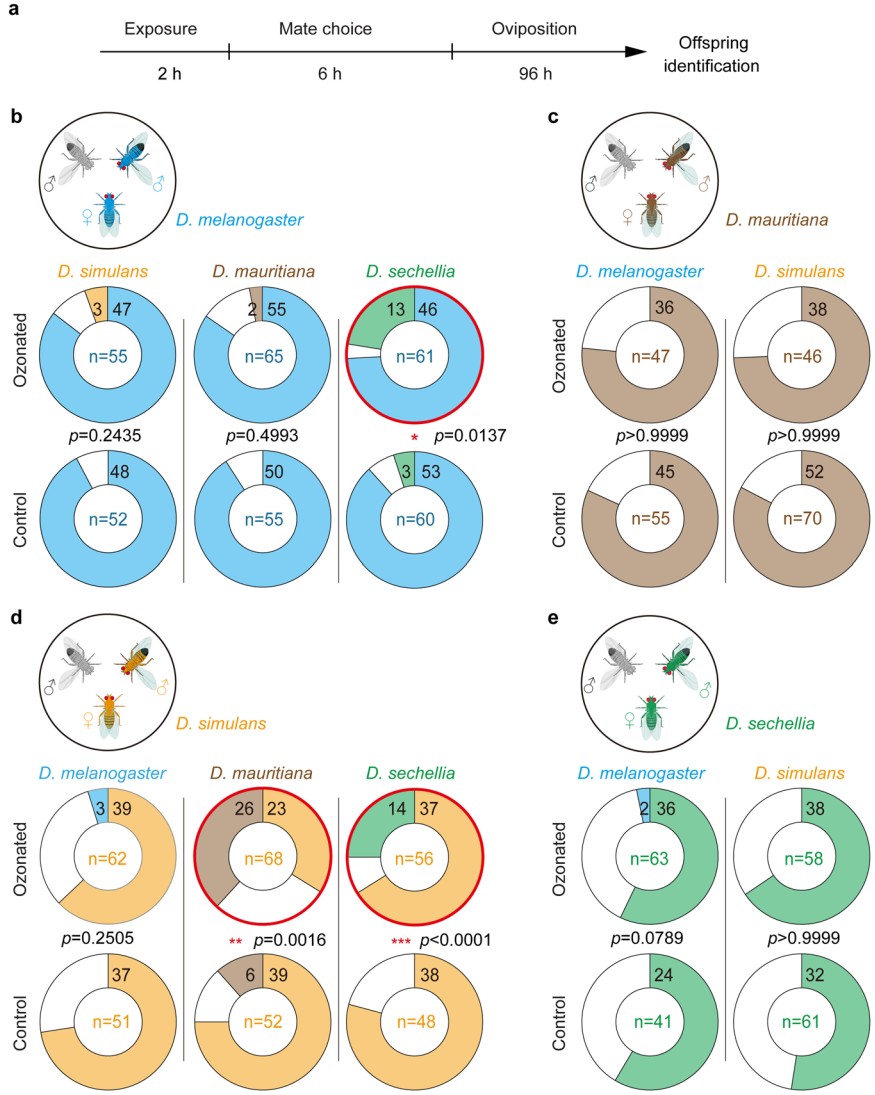

**Fig. 2 | Ozone exposure can induce hybridization among closely related *Drosophila* species. a** Time line of experiment. Ozonated and control flies are exposed for 2 h to 100 ppb and 5 ppb of ozone, respectively. Afterwards, individual female flies are confronted with one intra- and one interspecific male for six hours. The existence or absence of hybrid offspring informs about the succeeding male. Donut plots of success rates of ozonated (top) and control (bottom) conspecific and allospecific males courting *D. melanogaster* (**b**), *D. mauritiana* (**c**), *D. simulans* (**d**), or *D. sechellia* (**e**) females. Sample sizes are provided in donut centers. Numbers in segments depict numbers of successful males. White segments, no male mated the female. Red donuts depict significant increase of hybridization after exposure to high levels of ozone. Two-tailed Fisher's exact test, *p < 0.05; **p < 0.01; ***p < 0.001.

exposure resulted in a complete lack of preference for the conspecific male (Fig. 2d). Although the species boundary between *D. simulans* females and *D. mauritiana* males seems to be already less strict under control conditions (Fig. 2d), exposure to increased levels of ozone obviously has the potential to intensify hybridization between these two species. We conclude that ozone exposure can impact mate choice between closely related *Drosophila* species. Notably, the significant effects observed between *D. simulans* females and both *D. mauritiana* and *D. sechellia* males, as well as *D. melanogaster* females and *D. sechellia* males (Fig. 2b, d) were observed in species combinations that occur in overlapping habitats. Having shown that 100 ppb ozone can corrupt some of the species' boundaries (Fig. 2), we next asked if 50 ppb ozone (i.e. a concentration level that nowadays is already much more common in urban areas[32,33,48,49]) might be sufficient to induce similar effects also. However, as we did not observe any increased levels of hybridization under these conditions (Fig. S3) 50 ppb of ozone do not seem to induce hybridization. This corresponds well with recent findings, that more than 50 ppb of ozone are needed to corrupt sexual communication within *D. melanogaster*[20]. We conclude that severe levels of pollutants are needed to corrupt species boundaries in flies of the *D. melanogaster* species complex.

## Ozone-induced hybridization can contribute to insect decline and ongoing gene flow

Based on Haldane's rule, that after a hybridization event the heterogametic sex is more likely to be sterile[50], *Drosophila* male hybrids are usually sterile and female hybrids are fertile[36,42,47,51,52]. Therefore, hybridization may lead either to an evolutionary dead end or to ongoing gene flow, respectively. We, therefore, next investigated the mating competitiveness of hybrids as compared to their pure parental

species in a mating-choice assay (Fig. 3a, a_i). We found in mate choice experiments, where *D. melanogaster* flies had to choose between hybrid or conspecific mates, that hybrids between *D. melanogaster* and one of the other species were always significantly less successful, indicating that *D. melanogaster* prefers to mate with conspecifics rather than hybrids (Fig. 3b, b_i). Similarly reduced mating success of hybrids was found in many tested combinations (all data marked with black asterisks in Fig. 3). In these cases, the higher mating success rates of pure species may reduce further gene flow of hybrids within a population and, hence, maintain species boundaries in a long term. Interestingly, hybrid *D. sim-mau* males, *D. sim-sec* males, and *D. sim-mel* males exhibited a mating advantage compared to their rival *D. simulans* males, with more hybrids successfully mating with *D. simulans* females than their purebred competitors (Fig. 3d_i; here and thereafter hybrids are named first by the acronym of their mother and then by that of their father parent; for a chemical analysis of hybrid pheromone blends see Fig. S5). As hybrid males are sterile, mating with a hybrid male results in a loss of fitness for the female. Therefore, any ozone-induced increased number of such hybrid males in a population might contribute to the insect populations' decline. Furthermore, *D. mauritiana* and *D. simulans* males did not show any preference for their conspecifics over *D. sim-mau* and *D. sim-sec* females, respectively (Fig. 3c, d). Contrary to sterile male hybrids, female hybrids are fertile. Therefore, any lack of discrimination against hybrid females facilitates ongoing gene flow between species, which could result in the decline of the original population. Whether this could even result in new speciation events remains to be tested. Ongoing gene flow would become more likely if a male strongly prefers a hybrid female over a conspecific one, as we observed for hybrid *D. sim-sec* females over conspecific females in *D. sechellia* males (Fig. 3e). This raises the

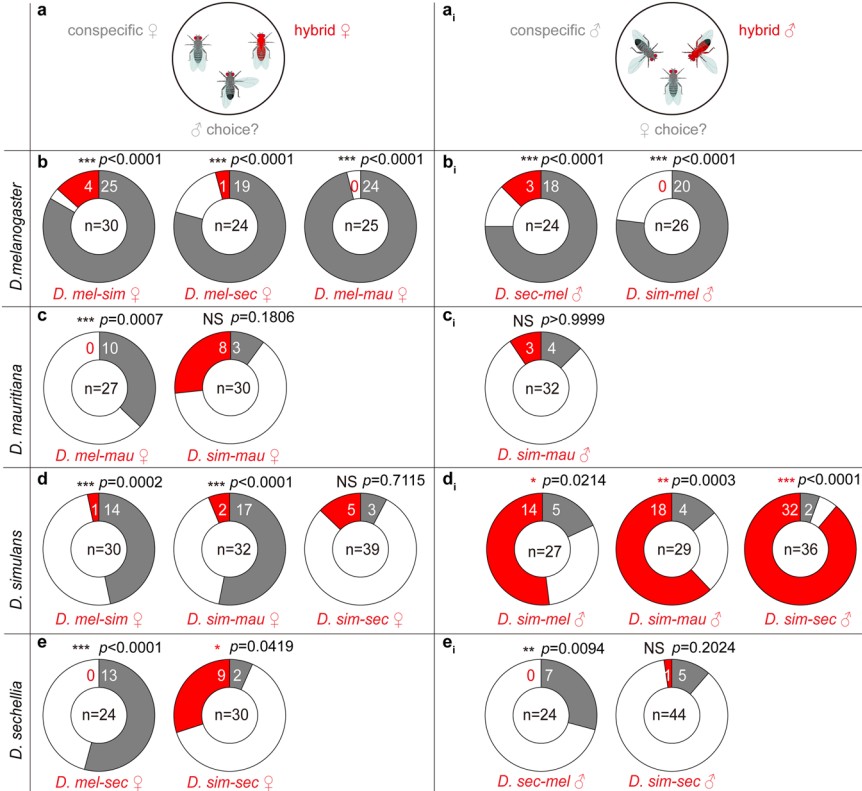

**Fig. 3 | Hybrid flies can exhibit mating advantages over pure species. a, ai,** Schematic drawing of the male-choice and female-choice assays where either males or females were exposed to a conspecific and a hybrid male and their choice was recorded during 1 h. Donut plots of mating choices of *D. melanogaster* (**b**), *D. mauritiana* (**c**), *D. simulans* (**d**), and *D. sechellia* (**e**) males and females (**b_i**–**e_i**) for conspecific (grey segments) or hybrid (red segments) mating partners. White segments, no mating within 1 h. Two-tailed *Fisher's exact* test, *p < 0.05; **p < 0.01; ***p < 0.001; black asterisks, conspecific preferred; red asterisks, hybrid preferred; NS no discrimination. Hybrids are named first by the acronym of their mother and then by that of their father parent.

possibility that ozone-induced hybridization events may result in an evolutionary dead end due to hybrid sterility and/or mating disadvantages of hybrids. However, some combinations between *D. simulans* and *D. mauritiana* as well as *D. simulans* and *D. secchellia* have the potential to result in continued gene flow over consecutive generations.

We next assessed the potential for ongoing gene flow by testing different parameters of the hybrids' fitness. Given that all male hybrids are known to be sterile[12,17,42,50] and that ozone-exposure especially induced the hybridization of *D. simulans* females with *D. sechellia*, and *D. mauritiana* males (Fig. 2), whose hybrid offspring turned out to be competitive in mating choice assays (Fig. 3) we here focused on those hybrids. When analyzing egg numbers, egg hatching rates, larval development time, and the development success rate from egg to adult, we did not find any dramatic hybrid inviabilities. The parameters revealed from offspring of *D. simulans* females that either mated with *D. sechellia* or *D. mauritiana* males and offspring of hybrid back crosses in most cases did not differ from that of pure *D. simulans* or *D. mauritiana* flies (but in some cases even outperformed those of pure *D. sechellia*) (Fig. S6). We conclude that via facilitating hybridization of *D. simulans* females with males of *D. mauritiana* and *D. sechellia*, ozone might induce long-lasting effects of gene flow in the corresponding sympatric populations.

Gene flow between the natural sympatric populations of *D. simulans* and *D. sechellia* has already been observed[37]. Our findings that hybrids of both species and backcrosses of those hybrids at least in some parameters turned out to be more viable than *D. sechellia* but not *D. simulans* flies might lead to higher success of these hybrids in *D. sechellia* populations and, hence, might explain the observed unidirectionality of gene flow from *D. simulans* to *D. sechellia* in their natural populations[37].

Furthermore, we used ozone for our experiments as its concentrations in many parts of the world have been shown to increase at least temporarily. A much more stable and therefore "reliable" oxidant pollutant is however, nitric oxides. Nitric oxides increase close to all industrial places that exhibit a high combustion of fuel and do not fluctuate as much as ozone levels[53]. Due to their even stronger oxidative power, nitric oxides can be expected to have even more detrimental effects on the flies' pheromone levels. However, as regulations for lab work with nitric oxides are so strict, we focused our work on the effect of ozone.

In *Drosophila*, as in most animals, sex-recognition and courtship are multimodal[54–56]. Male flies belonging to the *D. melanogaster* species complex are e.g. known to produce species-specific songs during courtship[57–59]. Therefore, one could expect that, despite corrupted pheromone communication after ozone exposure, species boundaries would exist due to such non-chemical courtship cues. However, despite the species-specificity of the male songs, females of most species from the *D. melanogaster* complex seem to become sexually excited also by heterospecific songs[60]. This might explain, why non-pheromonal cues in some of our experiments were not sufficient to maintain species boundaries, when ozone degraded the fly pheromones.

Oxidant pollutants such as ozone and nitric oxide have been demonstrated to affect animals across various communicative dimensions. They can disrupt chemical communication between plants and their pollinating insects, interfere with enemy orientation, and disturb intraspecies sexual communication[20,61–63]. Here, we provide evidence that ozone can break the mating boundaries in some *Drosophila* species by oxidizing their unsaturated pheromone compounds. Many resulting hybrids are sterile or face a disadvantage when competing with their purebred same sex for mates (Fig. 3)[64]. By increasing the occurrence of these hybrids, oxidizing pollutants like ozone might further reduce the insects' fitness and, hence contribute to the insect decline.

## Methods

### Drosophila lines and chemicals stocks
Wild-type flies of *D. simulans* (14021-0251.01), *D. mauritiana* (14021-0241.150), *D. sechellia* (14021-0248.07) in this study were obtained from the National Drosophila Species Stock Center (NDSSC), Cornell university (https://www.drosophilaspecies.com/). The wild-type *D. melanogaster* (Canton-S) was from the Hansson's lab. All flies were reared at 25 °C, 12 h Light:12 h Dark and 70% relative humidity. Before experiments, wild-type virgin flies were collected by using $CO_2$ anesthesia. Seven to ten-day-old virgin flies were used in the behavioral and chemistry tests. Care and treatment of all flies complied with all relevant ethical regulations. Chemical compounds of the 11-cis-Vaccenyl acetate (cVA), (Z)−7-Tricosene (7-T), (Z)−9-Tricosene (9-T), (Z)−7-Pentacosene (7-P), (Z)−7,11-heptacosadiene (7,11-HD), and (Z)−7,11-nonacosadiene (7,11-HD) were purchased in high purity from Sigma-Aldrich and Cayman Chemical.

### Ozone exposure system
The ozone exposure system has been described in a previous study[20]. Briefly, compressed ambient air was first humidified to 70% relative humidity and then used to produce control air. Control air comes from ambient air. Ozone-enriched air was made from the control air by an ozone generator (Aqua Medic, Germany) which could produce up to 100 mg ozone/h. Different levels of ozonated experimental air could be produced by mixing clean with ozone-enriched air in the mixing box. Ozonated experimental air was dynamically stored in a mix box (a 100 L Plexiglas container), from which air was continuously probed for the ozone monitor (BMT 932, BMT Messtechnik GmbH, Germany), while at the same time, each 0.2 l/min were led into the four 70 ml plastic vials containing the flies. As a control another set of four 70 ml vials were connected to the airflow of the control air.

### TDU GC-MS
Ten-day-old virgin flies of *D. melanogaster*, *D. simulans*, *D. mauritiana*, and *D. sechellia* were first exposed to 100 ppb ozone or control air with 2 h, and afterwards frozen at −20 °C for 30 min. Afterwards individual flies were placed in microvials of thermal desorption tubes (GERSTEL, Germany). To test chemical profiles of hybrids, we sometimes used younger flies (but never younger than 5 days), as some combinations resulted in weak hybrids that hardly would survive until the 10th day. As internal standard 0.5 μl of C10-Br ($10^{-3}$ dilution in hexane) was used. A GERSTEL MPS 2 XL multipurpose sampler transfers desorption tubes to the GERSTEL thermal desorption unit (GERSTEL, Germany). Samples were desorbed at 250 °C for 8 min, then trapped at −50 °C in the liner of a GERSTEL CIS 4 Cooled Injection System (with liquid nitrogen for cooling). The components were transferred to the GC column by heating the programmable temperature vaporizer injector at 12 °C/s up to 270 °C and then keeping the temperature for 5 min. The GC-MS (Agilent GC 7890 A fitted with an MS 5975C inert XL MSD unit; Agilent Technologies, USA) was equipped with an HP5 column (Agilent Technologies, USA). The temperature of the gas chromatograph oven was held at 50 °C for 3 min and then increased by 15 °C /min to 230 °C and then by 20 °C /min to 280 °C, held for 20 min. Mass spectra were taken in EI mode (at 70 eV) in the range from 33 m/z to 500 m/z.

### Posterior lobe of male genital dissection
To build the posterior lobe atlas, 20 males and 20 females from different species were forced to mate in a vial with standard fly food and the posterior lobes of the resulting male offspring (at the age of 7 days after eclosion) were dissected. All male flies including the pure and hybrids were 7-day-old virgins. During dissection, the flies were anesthetizing by $CO_2$. The posterior lobe was separated from male genital by using a scalpel and a needle. Posterior lobes were put on microscope slides (Paul Marienfeld GmbH & Co. KG, Germany), immersed by methyl salicylate (Sigma-Aldrich, Germany), and covered by a coverslip

(Carl Roth GmbH & Co. KG, Germany). On each slide, 15 posterior lobes from different individuals were placed. Slides were photographed using an AXIO Zoom V.16 (ZEISS, Germany, Oberkochen) with a 1.25× PlanApo Z objective (ZEISS, Germany, Oberkochen).

### Interspecies two-choice test

Both sexes of 10-day-old virgin flies from four species, *D. melanogaster*, *D. simulans*, *D. mauritiana*, and *D. sechellia*, were included in the experiment. The flies were exposed to 100 ppb ozone for 2 h (or in an additional experiment for 50 ppb for 2 h). Subsequently, a single female fly was introduced into each vial (15 mL volume) together with one conspecific and one allospecific male. The vials contained 2 mL of standard fly food. A control group was also established, where the flies were exposed to the ambient air for 2 h. To ensure that the female flies mated only once, males were removed from the vial after 6 h. Female flies were left in the vial to lay eggs. Once the first pupa appeared, the female flies were removed from the vials. In each vial, the sex ratio and the posterior lobe of at least three male flies was examined.

### Competition two-choice test

In the competition mating assays, both pure species and hybrid flies were marked by UV-fluorescent powder of different colors. Fluorescent powder purchased from Maxmax.com (https://maxmax.com; red: UVXPBR; blue: UVXPBB). During the competition two choice test, one hour was observed manually and mating success was recorded by identifying the successful rival under UV light. Ten-day-old hybrids and their pure-bred rivals were tested, except for the 5-day-old female hybrids *D. mel-sec* and female *D. mel-sim* (that often would not survive until their 10th day).

### Fitness of female hybrids and purebred species

To assess the fecundity of both purebred and hybrid females, we utilized 7-day-old virgin males and females. A single male and female were introduced into a 10 mL plastic vial containing 2 mL of standard fly food. Following a 24-h mating period, the male was removed, and the female was transferred to a new food vial daily from day 1 to day 5. Egg counts for each vial were conducted under a microscope after this timeframe. For evaluating egg hatch rates, 20 males and 20 females were placed together in plastic vials covered with a food petri dish (60 × 14 mm, Ratiolab Gmbh) as an egg pool. Females were permitted to lay eggs within a 24-h period. Subsequently, 10 eggs from each egg pool were transferred to a new food petri dish, and hatched eggs were counted after another 24 h. Finally, the total development time was measured as the duration from egg laying to the emergence of the first pupa.

### Statistical analyses

Statistical analyses (see the corresponding legends of each figure) and preliminary figures were conducted using GraphPad Prism v. 8 (GraphPad Software, USA). Figures were then processed with Adobe Illustrator CS5. All tests are two tailed. *Unpaired t* test (Fig. 1) and *Fisher's exact* test (Figs. 2, 3) were used.

### Reporting summary

Further information on research design is available in the Nature Portfolio Reporting Summary linked to this article.

## Data availability

All data are available in the main text or the supplementary materials. Source data are provided with this paper.

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

## Acknowledgements

We thank Prof. Richard Benton from the University of Lausanne for providing *D. mauritiana* flies. We also thank I. Alali and S. Trautheim for help with fly breeding. This research and all co-authors were supported through funding by the Max Planck Society and specifically through funding to the Max Planck Center "Next Generation Insect Chemical Ecology".

## Author contributions

N.J.J., B.S.H. and M.K. designed the research plan and N.J.J. performed most of the experiments. X.Q.D. performed several competition two-choice tests. D.V. designed and constructed the ozone exposure device and the courtship arenas. N.J.J. analyzed and quantified pheromone compounds. N.J.J. analyzed experimental data. N.J.J., B.S.H. and M.K. wrote the paper. All authors edited the manuscript.

## Funding

## Competing interests

The authors declare no competing interests.
