## [Peer Review File · Nature Communications]

Elevated ozone disrupts mating boundaries in drosophilid fliesReviewer #1 (Remarks to the Author):

Jiang et al. present a follow-up to their previous findings that exposure to high levels of ozone oxidizes cuticular pheromones of *Drosophila* (Jiang et al 2023 Nat. Com.). In their previous work, Jian et al showed that exposing males for 2h to 100 ppb ozone resulted in longer mating latencies and increased courting of other males. This previous paper ended by examining the effect of the same male treatment across 10 other *Drosophila* species and found largely consistent results. In their current submission, Jiang et al. ask how the same treatment (2h exposure to 100 ppb ozone) impacts inter-species mating behaviors. Using the closely-related *D. melanogaster*, *D. simulans*, *D. mauritiana*, and *D. sechellia* species, they provide evidence that:

1. The ozone treatment reduces the levels of the main cuticular hydrocarbons of all 4 species
2. The ozone treatment increased some inter-species matings
3. Hybrid offspring from the inter-species experiments resulted in *D. sim-mau* males and *D. sim-sec* males having greater mating success compared to pure *D. simulans* males. They also highlighted that both *D. mauritiana* and *D. simulans* males did not show any preference for their conspecifics over *D. sim-mau* and *D. sim-sec* females.

They conclude by arguing that their data highlights the ability of ozone to reduce conspecific matings and to break down species boundaries which can lead to both deleterious effects and potentially new species (in the case of the hybrids with putative fitness advantages).

Main Comments:

Overall, the question is very interesting and arise naturally from the authors' previous work. And though the results are intriguing, several important issues remain unaddressed and experimental design issues limit the study. In the end it is not clear that the current set of experiments fully supporting the conclusions that the authors draw. In addition, the paper would be strengthened by expanding the general framing of the question(s) in light of related previous published work.

- One wonders the temporal dynamics at play between spikes of high ozone and evolutionary processes. Further discussion and investigation of this question would strengthen this paper and the claims made by the authors as to the importance of ozone spikes and species boundaries. If high levels of ozone are relatively fleeting and the processes of speciation is likely to be much slower, how important is the effect expected to be? Additionally, most of the species are thought to have large population sizes with continuous breeding. How large would the effect of high ozone have to be in order to appreciably depress matings of conspecifics and thereby run the risk of population decline? Although it may be outside the area of the author's expertise, a model examining these parameters would go a long way in informing/shaping our intuition on this topic.

- Somewhat related to the above point, as noted in their 2023 Nat. Com. paper (Fig. S5), 5 days following the 2h to 100 ppb ozone treatment *D. melanogaster*'s cuticular hydrocarbons return to pretreatment levels. However, there were no experiments at intervals between 1 and 5 days. Additional resolution on this "return to normal" would be insightful. Additionally, it remains unclear if this same time frame exists for non-*D. melanogaster* species. It could be that other species take much longer to return to normal cuticular hydrocarbon levels (or much shorter). These parameters would also shape how we think about inter-species interactions.

- Regarding result #2 (above), although there does appear to be some increase in hybridization mating (as shown in Fig. 1), as only 3/10 experiments were significant additional caution should be placed on the interpretations. This is particularly the case given that relatively rare gene flow occurs between some of these species (e.g., 10.1371/journal.pgen.1007341 and the data shown by the authors).

- Have the authors carried out GC-MS on hybrids? If so, how would these results inform the observed matings?

- The broader context/background involving the processes/models of speciation was incomplete in general terms - in the framing of the narrative - and with respect to the species studied. For example, explicit discussion of the type of speciation models that the authors have in mind throughout the paper is never clearly made. It seems that sympatric speciation is often in mind, but this needs to be explicit if it is the case. The field of speciation biology is quite rich and the way these results fit within it should be expanded. With respect to these four species, considerable work has been carried out in trying to infer the speciation process among these species but few details of this work are referenced or discussed (for example, 10.1093/genetics/156.4.1913 , 10.1111/j.1558-5646.1989.tb04233.x , 10.1101/gr.130922.111 , 10.1371/journal.pgen.1007341, and later follow-up work). Other citations for the cuticular hydrocarbons across the four species would be useful as well. Additionally, there is also quite a lot of work on other behavioral differences involved in species recognition for these four species that do not involve pheromones, for example song preferences (e.g. the work of David Stern among others). How might these non-pheromone barriers to hybridization be involved in nature when ozone levels are temporarily high?

- The authors emphasize the possible importance of hybrid advantages (largely stemming from the results observed for *D. sim-mau* males and *D. sim-sec* males). These are strong claims as hybrids are usually at a disadvantage. When species are in sympatry, new species may arise if the fitness of the hybrid is greater relative to the non-hybrid species. To really substantiate this claim it would have been insightful to examine hybrid vs. hybrid interactions as well as testing for other life history traits that would indicate fitness advantages beyond the increased mating rates. For example, authors could compare rates of mating success from parental species to those from a "hybrid-by-hybrid" mating choice setup. From these experiment, the authors could measure basic traits like the number of laid eggs, the fraction of egg hatching, development time, and the fraction of individuals able to reach adulthood. These type of data are important for the claims of hybrid fitness. It would also be an opportunity to reveal potential post-zygotic isolation that, as much as pre-zygotic isolation, are important component during speciation processes.

- Regarding the authors' claim on lines 116-122, the citation used (#29) only pertains to *D. melanogaster*, yet the sentence implies the results apply to the four species. Is this a citation error? If it is not a citation error, then it would seem the certainty of the matings are drawn into question.

- Regarding the posterior lobe atlas, it seemed that the hybrids were the hardest to discriminate. Why have the authors not tested for hybrids genetically? Perhaps a compromise would be to demonstrate on a test set of genetically defined matings the accuracy of using the lobe atlas approach. Additionally, 10 combinations are shown in the Fig. S1 but shouldn't there be 12 - aren't the *Dmaur* - *Dmaur* vs *Dsech* and *Dsech* - *Dsech* vs. *Dmaur* combinations missing?

- Given the likely importance of mating latency between different species in the wild, it would have been useful to quantify this parameter as the authors did in their previous paper.

- Additional background on the species' distributions around the globe would be helpful. How might the fact that some of the species have subpopulations with varying degrees of gene flow while other species are likely more panmictic be impacted by temporary changes in ozone?

- The mating rates are quite low in Fig. 3. Additional discussion about why this is and the impact that it has on the results are needed.

- The sample sizes vary substantially the experiments summarized in Fig. 3. This results in varying statistical power to detect differences across experiments. Importantly, this likely limits the

comparative statements that the author make in the related text. Additional examination of this issue is needed.

Minor Comments:

- To increase comprehension, in Fig 2 the two rows of donut plots in panels B-E could be labeled with "ozonated" and "control".
- For Fig 1's t-test are the variance equal and data normally distributed?

Reviewer #2 (Remarks to the Author):

Reviewer #3 (Remarks to the Author):

In this manuscript the authors investigate the effects of ozone exposure on the mating and reproduction of several *Drosophila* species that are usually reproductively isolated. They show that exposure to 100ppb ozone increases the chance of hybridization in small scale experiments. They also show that reproductively viable hybrids are in certain cases able to successfully compete against conspecifics. This observation is potentially important and opens up several novel questions for future scientific investigation.

My perceived weakness in the current study is that only a high ozone concentration (100ppb) was used to represent a polluted environment. It would have been of value to assess the responses at a lower concentration in addition to this high level.

Aside from this observation, I find the paper to be mostly well written (there are a few sentences in need of editing in the materials and methods section), and very interesting. I have only minor suggestions to add.

Figure 1c: In the legend it refers to dark or bright plots, but it was not immediately evident what was referred to. It can be worked out, but I would suggest changing the description (or maybe the colors).

Figure 2: Mark the ozone exposed scenario more clearly in the figure.

Line 52: I suggest not starting this sentence with 'Obviously'.

Line 349: *D. putrida* is not subject to research in this manuscript so can be removed from the list.

Line 356: Check the nomenclature convention used and the spelling of 'cis vanccenyl acetate'.

REVIEWER COMMENTS

We thank all reviewers for the helpful comments. You will find below our answers in red and new text that was added to the manuscript in yellow. We have also color marked the next within the manuscript.

With best regards in the name of all coauthors,

Markus Knaden

Reviewer #1 (Remarks to the Author):

Jiang et al. present a follow-up to their previous findings that exposure to high levels of ozone oxidizes cuticular pheromones of *Drosophila* (Jiang et al 2023 Nat. Com.). In their previous work, Jian et al showed that exposing males for 2h to 100 ppb ozone resulted in longer mating latencies and increased courting of other males. This previous paper ended by examining the effect of the same male treatment across 10 other *Drosophila* species and found largely consistent results. In their current submission, Jiang et al. ask how the same treatment (2h exposure to 100 ppb ozone) impacts inter-species mating behaviors. Using the closely-related *D. melanogaster*, *D. simulans*, *D. mauritiana*, and *D. sechellia* species, they provide evidence that:

1. The ozone treatment reduces the levels of the main cuticular hydrocarbons of all 4 species
2. The ozone treatment increased some inter-species matings
3. Hybrid offspring from the inter-species experiments resulted in *D. sim-mau* males and *D. sim-sec* males having greater mating success compared to pure *D. simulans* males. They also highlighted that both *D. mauritiana* and *D. simulans* males did not show any preference for their conspecifics over *D. sim-mau* and *D. sim-sec* females.

They conclude by arguing that their data highlights the ability of ozone to reduce conspecific matings and to break down species boundaries which can lead to both deleterious effects and potentially new species (in the case of the hybrids with putative fitness advantages).

Main Comments:

Overall, the question is very interesting and arise naturally from the authors' previous work. And though the results are intriguing, several important issues remain unaddressed and experimental design issues limit the study. In the end it is not clear that the current set of experiments fully supporting the conclusions that the authors draw. In addition, the paper would be strengthened by expanding the general framing of the question(s) in light of related previous published work.

R: We thank the reviewer for his insightful and fair comments on our study. We have conducted the additional experiments/measurements the reviewer asked for and have added more background regarding the species' distribution and our ideas regarding ozone potentially paving the way for hybrid speciation.

Q1.- One wonders the temporal dynamics at play between spikes of high ozone and evolutionary processes. Further discussion and investigation of this question would strengthen this paper and the claims made by the authors as to the importance of ozone spikes and species boundaries. If high levels of ozone are relatively fleeting and the processes of speciation is likely to be much slower, how important is the effect expected to be? Additionally, most of the species are thought to have large population sizes with continuous breeding. How large would the effect of high ozone have to be to in order to appreciably depress matings of conspecifics and thereby run the risk of population decline? Although

it may be outside the area of the author's expertise, a model examining these parameters would go a long way in informing/shaping our intuition on this topic.

R: We thank the reviewer for raising this important question. Ozone concentrations indeed fluctuate over the year, but especially also during the days. In the manuscript we hypothesized that even short-term exposure to ozone could lead to long-term effects on species boundaries. As the reviewer criticizes in another comment, we did not test, how long the ozone-induced reduction of pheromones in the different species might last. We agree that this information is important to draw a conclusion, whether ozone might have long-term effects on species boundaries. We, therefore, conducted new experiments, where we measured pheromone levels of ozonated flies after 24 and 48 hours. While we still found significantly decreased levels of pheromones after 24 hours, the normal pheromone levels were reestablished after 48 hours (new Figure S1). As peaks of ozone concentrations often re-occur on a daily base (Raga and Raga, 1999), flies in these areas might not be able to re-establish their pheromone levels in-between. Therefore, even temporarily restricted peaks of high ozone can potentially induce long-lasting effects on *Drosophila* mating boundaries. We agree that it is difficult to judge, whether this can lead to a population decline. We have added to the manuscript:

“Analyzing the recovery rates of pheromone levels after ozone exposure resulted in still significant differences after 1 day, that, however, disappeared after 2 days (Fig. S1). Hence, it takes at least 2 days for the pheromone levels to recover. As peaks of ozone concentrations often re-occur on a daily base²⁶, flies in polluted areas might not be able to re-establish their pheromone levels in-between.”

We have also added a new supplementary figure with the recovery rates of pheromones after ozone exposure:

Fig S1. Quantitative analysis of cVA and pheromonal CHCs after ozone exposure recovery in four *Drosophila* species. **a**, Time line of experiment. Ozonated and control flies are exposed for two hours to 100 ppb ozone and ambient air, respectively. After that flies were placed into food vial and let them recover 24h or 48h. **b**, Quantitative analysis after 24h recovery. **c**, Quantitative analysis after 48h recovery. Data present as min to max. Unpaired *t*-test. * $p < 0.05$; ** $p < 0.01$; *** $p < 0.001$; NS, no significant difference.

We use the species of the *Drosophila melanogaster* complex, because both their pheromone blends and their sexual behavior are well established. Within the *Drosophila* genus, however, many more sympatric species pairs exist that can hybridize (Coyne and Orr 1989) and whose species boundaries therefore might also become affected by increased levels of ozone. Furthermore, we used ozone for our experiments as its concentrations in some parts of the world have been shown to increase at least temporarily. A much more stable and therefore “reliable” oxidant pollutant is however, nitric oxides. Nitric oxides increase close to all

industrial places that exhibit a high combustion of fuel and do not fluctuate as much as ozone levels. Nitric oxides due to their strong oxidative power, can be expected to have even more detrimental effects on the flies' pheromone levels. However, as regulations for lab work with nitric oxides are so strict, we focused our work on the effect of ozone. We have added to the manuscript:

“In our experiments we used the species of the *Drosophila melanogaster* complex, because both their pheromone blends and their sexual behavior are well established. Within the *Drosophila* genus, however, many more sympatric species pairs exist that can hybridize⁶⁵ and whose species boundaries therefore might also become affected by increased levels of ozone. Furthermore, we used ozone for our experiments as its concentrations in many parts of the world have been shown to increase at least temporarily. A much more stable and therefore “reliable” oxidant pollutant is however, nitric oxides. Nitric oxides increase close to all industrial places that exhibit a high combustion of fuel and do not fluctuate as much as ozone levels⁷⁹. Due to their even stronger oxidative power, nitric oxides can be expected to have even more detrimental effects on the flies' pheromone levels. However, as regulations for lab work with nitric oxides are so strict, we focused our work on the effect of ozone.”

In recent decades, there has been a notable decline in insect populations, with a particular focus on species such as honey bees, beetles, and others, extensively examined in numerous studies (Cardoso et al., 2020; Zhou et al., 2023). Factors such as habitat loss, the widespread use of chemical pesticides, the escalating climate crisis, among others, are identified as the primary contributors (Wagner et al., 2021).

In our recent paper (Jiang et al. 2023) we showed that increased levels of ozone result in increased intraspecific mating latencies (as males exhibit lower levels of pheromones and, hence, become less attractive to females). This suggests a less efficient use of energy and exposes individuals to higher predation risks. In our current manuscript we show that exposure to oxidant pollutants can furthermore

result in increased hybridization and, hence, the production of often sterile offspring. Based on all these findings we argue, that oxidant pollutants might contribute to the insect decline. Our hypothesis is supported by a comprehensive analysis, incorporating both chemical and behavioral evidence. We are, however, well aware, that it is difficult to test whether and how much pollutants really contribute to the insect decline. Although modeling analysis falls outside our expertise, we feel that we present a robust case supported by evidence at both the chemical and behavioral levels.

Q2.- Somewhat related to the above point, as noted in their 2023 Nat. Com. paper (Fig. S5), 5 days following the 2h to 100 ppb ozone treatment D. melanogaster's cuticular hydrocarbons return to pretreatment levels. However, there were no experiments at intervals between 1 and 5 days. Additional resolution on this "return to normal" would be insightful. Additionally, it remains unclear if this same time frame exists for non-D. melanogaster species. It could be that other species take much longer to return to normal cuticular hydrocarbon levels (or much shorter). These parameters would also shape how we think about inter-species interactions.

R: We thank the reviewer for this comment, followed his suggestion, and conducted additional TDU GC-MS tests to assess how long it takes after ozone exposure until original pheromone levels are re-established (see new supplemental figure and added text mentioned above).

Q3- Regarding result #2 (above), although there does appear to be some increase in hybridization mating (as shown in Fig. 1), as only 3/10 experiments were significant additional caution should be placed on the interpretations. This is particularly the case given that relatively rare gene flow occurs between some of these species (e.g., 10.1371/journal.pgen.1007341 and the data shown by the authors).

R: We thank the reviewer for pointing us at Schrider et al. 2018, which indeed is very relevant for our manuscript. We now cite this study and incorporated its insights into our interpretations throughout the text. Our main conclusion is that hybridization following ozone exposure primarily occurred in the pairings of female *D. melanogaster* with male *D. sechellia*, female *D. simulans* with male *D. sechellia*, and female *D. simulans* with male *D. mauritiana* (see Figure 2). Notably, our study demonstrates that *D. simulans* females can interbreed with *D. sechellia* males, and the resulting hybrid, *D. sim-sec*, was not discriminated against *D. simulans* males and was even preferred over *D. sechellia* males (refer to Figure 3). This finding suggests a potential gene flow between *D. simulans* to *D. sechellia*, aligning with the conclusions drawn in the cited references (Schrider et al., 2018; Matute and Ayroles, 2014), that might be further reinforced by oxidant pollutants. While our behavioral data may not comprehensively address questions related to gene flow on a population level, it contributes insights, under which conditions (increased air pollutants) gene flow potentially can be expected to increase. We have now also included additional experiments on hybrid fitness that might partially explain, why gene flow between *D. simulans* and *D. sechellia* was observed to be one-directional (i.e. from *D. simulans* towards *D. sechellia*) in Schrider et al. 2018. Backcrosses of *D. simulans-sechellia* females with either *D. simulans* males laid less eggs and their offspring exhibited a longer development time than the purebred *D. simulans*, while the backcrosses of hybrid females with *D. sechellia* males laid more eggs and exhibited a higher survival rate than purebred *D. sechellia* flies (new Figure S6). The differences in fitness parameters might explain, why hybrids might successfully contribute their genes to *D. sechellia* populations but less well to *D. simulans* populations, i.e. why the gene flow was observed to be one-directional (Schrider et al., 2018). In addition, one-directional gene flow could also be due to hybrid *D. simulans-sechellia* being potentially attracted to the *D. sechellia* host (noni fruit). In that case hybrids would rather meet and mate with *D. sechellia* than with *D. simulans*, that show up at noni less frequently (Matute and Ayroles, 2014). However, as testing hybrids for host preferences was beyond the scope of our study, we cannot comment on this.

We have added to the manuscript:

“Gene flow between the natural sympatric populations of *D. simulans* and *D. sechellia* has already been observed³⁷. Our findings that hybrids of both species and backcrosses of those hybrids at least in some parameters turned out to be more viable than *D. sechellia* but not *D. simulans* flies might lead to higher success of these hybrids in *D. sechellia* populations and, hence, might explain the observed unidirectionality of gene flow from *D. simulans* to *D. sechellia* in their natural populations³⁷.”

Q4- Have the authors carried out GC-MS on hybrids? If so, how would these results inform the observed matings?

R: We agree that the pheromone signature of hybrids is an interesting point. We have now carried out TDU GC-MS experiments with all those hybrids that we tested for mating competitiveness (new Figure S5). While some of the hybrid pheromone patterns correspond well with the observed behavior (e.g. *D. mauritiana* males mate similarly often with *D. sim-mau* females v.s. *D. mauritiana* females, which also share the same pheromone amounts), others do not (e.g. *D. melanogaster* males mate more often with *D. melanogaster* females than with *D. mel-sec* hybrids, although both females share the same pheromones). Obviously other parameters (e.g. the females' acceptance of the male song) play additional roles here. However, although the inheritance of mating signals like pheromones and acoustic courtship in hybrids is an interesting topic, the main goal of our courtship experiments with hybrids was to show, whether pollution-induced hybrids are an evolutionary dead-end or potentially can transfer their genes further. Following a later suggestion of the same reviewer, we have now added experiments regarding the fertility of the hybrids (see below).

We have added to the result part: “for a chemical analysis of hybrid pheromone blends see Fig. S5” and to the supplements: “While some of the hybrid pheromone patterns correspond well with the observed behavior (e.g. *D. mauritiana* males

mate similarly often with *D. sim-mau* females v.s. *D. mauritiana* females, which also share the same pheromone amounts), others do not (e.g. *D. melanogaster* males mate more often with *D. melanogaster* females than with *D. mel-sec* hybrids, although both females share the same pheromones). Obviously other parameters (e.g. the females' acceptance of the male song) play additional roles here."

Q5- The broader context/background involving the processes/models of speciation was incomplete in general terms - in the framing of the narrative - and with respect to the species studied. For example, explicit discussion of the type of speciation models that the authors have in mind throughout the paper is never clearly made. It seems that sympatric speciation is often in mind, but this needs to be explicit if it is the case. The field of speciation biology is quite rich and the way these results fit within it should be expanded. With respect to these four species, considerable work has been carried out in trying to infer the speciation process among these species but few details of this work are referenced or discussed (for example, 10.1093/genetics.tb04233.x , 10.1101/gr/journal.pgen.1007341, and later follow-up work). Other citations for the cuticular hydrocarbons across the four species would be useful as well.

R: We apologize for the unintentional lack of relevant citations. In the new version, we have placed a stronger emphasis on the speciation process and population genetics. We have now included the suggested references and have added to the discussion:

"Reproductive isolation i.e. the lack of gene flow between populations is regarded as an important driver of speciation (reviewed by Mallet, 2006)⁵³. Such reproductive isolation often is a result of geographic isolation of so-called allopatric populations that via different selective pressures or genetic drift become more and more dissimilar and finally speciate. In addition, few examples of sympatric speciation (i.e. the evolution of a new species in close proximity of its ancestral

species) have been identified in e.g. African cichlids⁵⁴⁻⁵⁶ or the apple maggot fly⁵⁷⁻⁵⁹. Finally, some species seem to be the result of hybrid speciation⁶⁰, where the hybridization between closely related species finally results in the evolution of a new species. The most prominent insect example is the species-rich genus of *Heliconius* butterflies⁶¹, where hybridization of two closely related species can result in a fertile hybrid that by its wing pattern and behavior is reproductively isolated from the two donor species⁶². Similarly, there is one reported case of hybrid speciation for *Drosophila*, where hybrids of *D. ananassae* and *D. parapallidosa* obviously evolved into the new species *D. cf. parapallidosa*⁶³. In our case *D. sechellia*, and *D. mauritiana* most probably have evolved from a large mainland population of a shared ancestor with *D. simulans* through allopatric speciation based on two island colonization events⁶⁴. *D. simulans*, like *D. melanogaster* nowadays is globally distributed and also occurs on Mauritius and the Seychelles, i.e. the islands originally inhabited by *D. mauritiana* and *D. sechellia*. It has been shown that in *Drosophila* flies during speciation usually first prezygotic isolation (i.e. via courtship and mating boundaries) and afterward postzygotic isolation (via hybrid sterility and inviability) are established^{65,66}. *D. simulans*, *D. sechellia*, and *D. mauritiana* belong to the *simulans* species complex and have established prezygotic isolation based on e.g. species-specific pheromonal blends^{18,67-69} and courtship songs⁷⁰⁻⁷². Their post-zygotic isolation, however, is incomplete, as only male hybrids are sterile, while female hybrids are fertile. On both islands, gene flow via hybridization events between *D. simulans* and its close relatives has been reported (with *D. sechellia*³⁶; with *D. mauritiana*⁵²), suggesting that presynaptic isolation between these species is not absolute. Our data reveal, that oxidant pollutants like ozone have the potential to corrupt prezygotic isolation and, hence, make hybridization events more likely. As at least some of the resulting hybrids seem to be competitive regarding mate choice (Fig. 3) and reproduction (Fig. S6), such hybridization events potentially could lead to hybrid speciation.”

Q5 (2)- Additionally, there is also quite a lot of work on other behavioral differences involved in species recognition for these four species that do not

involve pheromones, for example song preferences (e.g. the work of David Stern among others). How might these non-pheromone barriers to hybridization be involved in nature when ozone levels are temporarily high?

R: We thank the reviewer for this comment and have added to the discussion:

"In *Drosophila*, as in most animals, sex-recognition and courtship is multimodal⁷³⁻⁷⁵. Male flies belonging to the *D. melanogaster* species complex are e.g. known to produce species-specific songs during courtship⁷⁶⁻⁷⁸. One, therefore, could have expected, that despite corrupted pheromone communication after ozone exposure, species boundaries would exist due to such non-chemical courtship cues. However, despite the species-specificity of the male songs, females of most species from the *D. melanogaster* complex seem to become sexually excited also by heterospecific songs⁷¹. This might explain, why non-pheromonal cues in some of our experiments were not sufficient to maintain species boundaries, when ozone degraded the fly pheromones."

Q6- The authors emphasize the possible importance of hybrid advantages (largely stemming from the results observed for *D. sim-mau* males and *D. sim-sec* males). These are strong claims as hybrids are usually at a disadvantage. When species are in sympatry, new species may arise if the fitness of the hybrid is greater relative to the non-hybrid species. To really substantiate this claim it would have been insightful to examine hybrid vs. hybrid interactions as well as testing for other life history traits that would indicate fitness advantages beyond the increased mating rates. For example, authors could compare rates of mating success from parental species to those from a "hybrid-by-hybrid" mating choice setup. From these experiment, the authors could measure basic traits like the number of laid eggs, the fraction of egg hatching, development time, and the fraction of individuals able to reach adulthood. These type of data are important for the claims of hybrid fitness. It would also be an opportunity to reveal potential

post-zygotic isolation that, as much as pre-zygotic isolation, are important component during speciation processes.

R: We fully endorse the proposition that the enhanced fitness of hybrids may contribute to speciation. In response to the reviewer's suggestions, we conducted a series of experiments to assess the fitness of hybrids. We added a new figure S6.

Figure S6. Fitness of female hybrid and purebred fly in laid eggs, hatch rate, development time, and the fraction from egg to adults. a, Laid egg numbers of each female in 5 days after mated. Figure shows mean ± SD. b, Egg hatch rate after 48h. c, Development time (days) from egg to pupa. Figure shows mean ± SD. d, Fraction of egg to adult. The x-axis shows combination of female/male. One-way ANOVA with Tukey's multiple comparisons test for a and c. Chi-square test with Bonferroni adjustment for b and d. Stars or characters with orange, green, and brown are showed the comparison with *D. sim*, *D. sec*. and *D. mau*, respectively. * $p < 0.05$; ** $p < 0.01$; * $p < 0.001$; NS, no significant difference.**

and have added to the manuscript:

“We next assessed the potential for ongoing gene flow by testing different parameters of the hybrids’ fitness. Given that all male hybrids are known to be sterile^{13,18,42,50} and that ozone-exposure especially induced the hybridization of *D. simulans* females with *D. sechellia* and *D. mauritiana* males (Fig. 2), whose hybrid offspring turned out to be competitive in mating choice assays (Fig. 3) we here focused on those hybrids. When analyzing egg numbers, egg hatching rates, larval development time, and the development success rate from egg to adult, we did not find any dramatic hybrid inviabilities. The parameters revealed from offspring of *D. simulans* females that either mated with *D. sechellia* or *D. mauritiana* males and offspring of hybrid back crosses in most cases did not differ from that of pure *D. simulans* or *D. mauritiana* flies (but in some cases even outperformed those of pure *D. sechellia*) (Fig. S6). We conclude that via facilitating hybridization of *D. simulans* females with males of *D. mauritiana* and *D. sechellia*, ozone might induce long-lasting effects of gene flow in the corresponding sympatric populations.”

Q7- Regarding the authors’ claim on lines 116-122, the citation used (#29) only pertains to *D. melanogaster*, yet the sentence implies the results apply to the four species. Is this a citation error? If it is not a citation error, then it would seem the certainty of the matings are drawn into question.

R: We added relevant citations and performed an experiment to observe the mating times of female specimens over a 6-hour period. Specifically, one female, accompanied by two males, was placed into a food vial, and their interactions were observed for the entire 6-hour duration. The results, showed in new Figure S2, indicate that within this time frame, the majority of females from the three tested species mated only once, while some females did not engage in any mating activity, and none of the females mated more than once. We have added to the text:

“As females of these four species do not re-mate within 6 hours after mating (*D. melanogaster*^{40,41}; *D. simulans*⁴¹, *D. sechellia*, and *D. mauritiana*, see Fig. S2), the identification of offspring informed us about the identity of the successful male (Fig. 2a).” and have added to the supplements:

Fig S2. Mating frequency during 6 hours when a female is confronted with two conspecific males. The donut plot shows numbers of flies that mated once (filled part of donut) or not at all (open part of donut). No fly was observed to mate more than once. *D. sim*, *D. simulans*; *D. sec*, *D. sechellia*; *D. mau*, *D. mauritiana*.

Q8- Regarding the posterior lobe atlas, it seemed that the hybrids were the hardest to discriminate. Why have the authors not tested for hybrids genetically? Perhaps a compromise would be to demonstrate on a test set of genetically defined matings the accuracy of using the lobe atlas approach. Additionally, 10 combinations are shown in the Fig. S1 but shouldn't there be 12 – aren't the Dmaur - Dmaur vs Dsech and Dsech - Dsech vs. Dmaur combinations missing?

R: Given that conserved mitochondrial genes are maternally inherited and unable to provide detailed information about hybridization, we would have had to focus on nuclear genes as potential candidates. However, our method, involving the testing of at least 3 males in each vial, would have had resulted in a substantial number of samples—over 3000—across approximately 50 replicates displayed in the 20 donut plots in Figure 2.

In contrast, we found that examining the posterior lobe morphology, a feature previously demonstrated as distinguishable and stable in numerous studies

(Matute and Ayroles, 2014; Markow and O'Grady, 2000), proved to be a more efficient approach in our case. For example, comparing the male posterior lobes from hybrids of *D. simulans-melanogaster*, *D. simulans-sechellia*, *D. simulans-mauritiana* to the parental males of *D. simulans* revealed distinct shapes, and a similar distinction was observed when comparing the genitalia of *D. sechellia-melanogaster* hybrids to those of *D. sechellia* males. The differences in posterior lobe morphology were easily observable under a stereoscope. We therefore think, that focusing on a morphological comparison, which is well established for *Drosophila* hybrids (Coyne 1983, Matute and Ayroles, 2014; Markow and O'Grady, 2000), we receive the important information in a more straightforward manner.

We reorganized Figure S3 and introduced an additional form to illustrate the hybridization results among the four *Drosophila* species. In the posterior lobe atlas, we focused on those hybrid male genitalia that we gained from forced crossings. Contrary to Coyne 1983 and Lachaise et al. 1986, but in line with Stamenkovic-Radak et al. 2009 we did not obtain any crossings from female *D. mauritiana* and male *D. simulans*. We therefore compared the genitalia of all males from our choice assay of *D. mauritiana* females with conspecific vs. *D. simulans* males with male *D. mauritiana* genitalia and the published hybrid genitalia (Coyne 1983). As all genitalia looked like *D. mauritiana* ones we conclude that *D. mauritiana* females also in the choice assay never mated with the *D. simulans* males. Furthermore, we did not include any *D. mauritiana/D.sechellia* crossings, as these two species inhabit different islands and do not exhibit any sympatric populations (We have clarified this now in the figure legend).

Fig S3. A hybridization overview between four *Drosophila* species and male posterior lobe morphology of *Drosophila* purebred species and hybrids. a, Hybridizations between four *Drosophila* species. b, Morphology of male posterior lobes. *D. sim*: *D. simulans*; *D. sec*: *D. sechellia*; *D. mau*: *D. mauritiana*. All hybrids are F₁ and named as F₀ female x F₀ male, e.g. *D. sim-mel* is the hybrid of a female *D. sim* and a male *D. mel*. Rep. 1- Rep. 3 indicate three example replicates. Any potential crossings from *D. mauritiana* and *D. sechellia* are excluded, as these species never meet in nature.

We have added to the results part:

“To identify the offspring from the above experiments, we, hence, performed no-choice assays between all species that potentially can hybridize in nature, i.e. we excluded crossings of *D. mauritiana* and *D. sechellia* that inhabit different islands and do not exhibit any sympatric population. From the pure species and the gained hybrid males, we constructed a posterior lobe atlas (Fig. S3). Contrary to Coyne (1983)⁴⁴ and Lachaise et al. (1986)⁴² but in line Stamenkovic-Radak et al. (2009)⁴⁷ we did not obtain any hybrids from no-choice assays using *D. mauritiana* females

with *D. simulans* males. Nevertheless, we compared the genitalia of males obtained from choice assays between both species with the hybrid morphology reported by Coyne 1983⁴⁴ and to the purebred *D. mauritiana*.”

Q9- Given the likely importance of mating latency between different species in the wild, it would have been useful to quantify this parameter as the authors did in their previous paper.

R: In our previous work, the majority of behavioral tests involved no-choice scenarios, allowing for recording through cameras and enabling the precise measurement of mating latency. However, in the current study, the focus shifted to two-choice tests aimed at illustrating the mating potential between different species. To distinguish fly species, fluorescent dye was employed as a marker, and its color could be visually observed by eye under ultraviolet (UV) light using a UV flashlight. Technically, discerning color details in videos posed challenges due to the need for high resolution, and fluorescence decayed under prolonged illumination. Moreover, flies marked with fluorescent dye exhibited grooming behavior, further diminishing fluorescence. We therefore decided to use a paradigm, where we did not observe the flies directly, but measured the mating success of the different males or females by screening the offspring for hybrids. This paradigm, however, hindered us from measuring latencies.

Q10- Additional background on the species' distributions around the globe would be helpful. How might the fact that some of the species have subpopulations with varying degrees of geneflow while other species are likely more panmictic be impacted by temporary changes in ozone?

R: We thank the reviewer for this question, and we have added more background information about these species to the manuscript. “In our case *D. sechellia*, and *D. mauritiana* most probably have evolved from a large mainland population of a

shared ancestor with *D. simulans* through allopatric speciation based on two island colonization events⁶⁴. *D. simulans*, like *D. melanogaster* nowadays is globally distributed and also occurs on Mauritius and the Seychelles, i.e. the islands originally inhabited by *D. mauritiana* and *D. sechellia*. It has been shown that in *Drosophila* flies during speciation usually first prezygotic isolation (i.e. via courtship and mating boundaries) and afterward postzygotic isolation (via hybrid sterility and inviability) are established^{65,66}. *D. simulans*, *D. sechellia*, and *D. mauritiana* belong to the *simulans* species complex and have established prezygotic isolation based on e.g. species-specific pheromonal blends^{18,67-69} and courtship songs⁷⁰⁻⁷². Their post-zygotic isolation, however, is incomplete, as only male hybrids are sterile, while female hybrids are fertile. On both islands, gene flow via hybridization events between *D. simulans* and its close relatives has been reported (with *D. sechellia*³⁶; with *D. mauritiana*⁵²), suggesting that presynaptic isolation between these species is not absolute. Our data reveal, that oxidant pollutants like ozone have the potential to corrupt prezygotic isolation and, hence, make hybridization events more likely.”

Q11- The mating rates are quite low in Fig. 3. Additional discussion about why this is and the impact that it has on the results are needed.

R: In *Drosophila* species, females usually determine, whether copulation occurs or not, while males are less choosy and often court heterospecific females. Despite often low mating rates, we observed male courtship behavior in all experiments, indicating that males showed willingness to mate. The lowest mating rates we observed in experiments with female *D. mauritiana* or *D. sechellia*. Food odors are known to increase mating motivation (Lin et al., 2015; Gorter et al., 2011; Gorter et al., 2016; Lebreton et al., 2012, Das et al. 2017) and we potentially could have increased the mating rates by mating the flies on food. However, as different food odors might affect the different fly species differently, we decided to stick to our

standard courtship method where three flies were put together in the absence of food.

Q12- The sample sizes vary substantially the experiments summarized in Fig. 3. This results in varying statistical power to detect differences across experiments. Importantly, this likely limits the comparative statements that the author make in the related text. Additional examination of this issue is needed.

R: We agree that different sample sizes might affect the conclusions that can be drawn and therefore performed more experiments have added more replicates to the Figure 3. However, despite of adding those, the conclusions remained similar.

Minor Comments:

- To increase comprehension, in Fig 2 the two rows of donut plots in panels B-E could be labeled with “ozonated” and “control”.

R: We marked the Fig 2 donuts plots with “Control” and “Ozonated”. Please see the Fig 2.

- For Fig 1’s t-test are the variance equal and data normally distributed?

R: We rechecked our data again very carefully. Most of the data are normally distributed and also for the other data we do not expect any e.g. bimodal distribution. We therefore tested all data with parametric statistics. However, we have now analyzed the data with a non-parametric test that, however, resulted in the same conclusion that pheromone levels significantly decrease after ozone exposure.

Reviewer #2 (Remarks to the Author):

Reviewer #3 (Remarks to the Author):

In this manuscript the authors investigate the effects of ozone exposure on the mating and reproduction of several *Drosophila* species that are usually reproductively isolated. They show that exposure to 100ppb ozone increases the chance of hybridization in small scale experiments. They also show that reproductively viable hybrids are in certain cases able to successfully compete against conspecifics. This observation is potentially important and opens up several novel questions for future scientific investigation.

My perceived weakness in the current study is that only a high ozone concentration (100ppb) was used to represent a polluted environment. It would have been of value to assess the responses at a lower concentration in addition to this high level.

R: We thank the reviewer for this comment and have added experiments with a lower concentration of 50 ppb ozone. We find that at this level, hybridization rates do not increase (new Fig. S2 (see below)). However, as local ozone peaks can already reach up to 250 ppb and rather frequently exceed 100 ppb in many places of the world, we still find that our results from exposure to 100 ppb are relevant.

Figure S4. Ozone exposure to 50ppb ozone is not enough to induce hybridization among closely related *Drosophila* species. Individual female flies are confronted with one intra- and one interspecific male for six hours. The existence or absence of hybrid offspring informs about the succeeding male. Donut plots of success rates of ozonated (middle) and control (bottom) conspecific and allospecific males courting *D. melanogaster* or *D. simulans* females. Sample sizes are provided in donut centers. Numbers in segments depict numbers of successful males. White segments, no mating occurred. Two-tailed Fisher's exact test.

Aside from this observation, I find the paper to be mostly well written (there are a few sentences in need of editing in the materials and methods section), and very interesting. I have only minor suggestions to add.

Figure 1c: In the legend it refers to dark or bright plots, but it was not immediately evident what was referred to. It can be worked out, but I would suggest changing the description (or maybe the colors).

R: We deleted the "dark and bright" in the figure legend. And we also changed the color for the Fig 1.

Figure 2: Mark the ozone exposed scenario more clearly in the figure.

R: We marked the Fig 2 donuts plots with “Control” and “Ozonated” in the left side. Please see the Fig 2.

Line 52: I suggest not starting this sentence with ‘Obviously’.

R: We rephrased this sentence and deleted ‘Obviously’.

Line 349: *D. putrida* is not subject to research in this manuscript so can be removed from the list.

R: We apologize and deleted the *D. putrida*.

Line 356: Check the nomenclature convention used and the spelling of ‘cis vanccenyl acetate’.

R: We checked the nomenclature convention in the whole manuscript and changed the ‘cis vanccenyl acetate’ to the ‘11-cis-Vanccenyl acetate’.

Reviewer #1 (Remarks to the Author):

We appreciate the efforts that the authors put into several sets of new experiments and textual modifications. Parts of this manuscript were strengthened as a result. However, two central and related aspects to this work remain incompletely addressed resulting in significant limitations to the study's main results.

The first is the claim that the timeframe over which the ozone treatment depletes fly pheromones is relevant at the timescale of speciation processes. We had previously encouraged the authors to better quantify the "recovery time" for pheromone profiles to return to pre-exposure levels and to consider how these temporal and population/population genetic processes would play out. We appreciate the additional experiments that were carried out to better define the recovery time for *D. melanogaster* and the other species. The result of these experiments indicate that the flies recover their pheromones quicker than previously suggested - two days instead of five. Though this may be a minor change (but in a direction that suggest a reduced role for ozone impacting pheromones and species boundaries), a significant limitation is that only speculations can be made on this point without a more general quantitative framework to guide our thinking and intuition. The authors have only added verbal arguments in favor of ozone's importance and have not provided additional quantitative arguments for how the process(es) would play out. As a result, the relevance of the ozone exposure on between-species dynamics is unconvincing.

The other major limitation in the current draft is that the authors still seem largely unaware of the relevant literature on speciation genetics directly relevant to questions they raise, including numerous studies that have investigated the genetic reasons that species within the *D. simulans* complex are unable to create new 'hybrid species' (as well as the broader literature on the genetics of speciation). In addition to the observation that the species conform to Haldane's rule, specific genetic incompatibles have been described (see citations below). These genetic incompatibles mean that even if the flies' pheromone-based recognition systems are altered other genetic barriers would prevent hybrid species from forming. It was also unclear if the authors realize that hybrids with sterile males would only result in gene flow through hybrid:parental species matings which would be unlikely to persist over long timespans given the many more conspecific matings. There are no reasons to suspect that these instances of gene flow would result in new species or the decline of a species in the way the authors suggest. Though the authors provided a considerable amount of new text on the topic of speciation, the hope was that it would have been used to better explain their own thoughts about how their observations/hypothesis in light of these speciation genetic papers and not in such general terms. As the manuscript stands, the conclusions and discussion are naive and too disconnected to the field that they aim to contribute to.

Several relevant speciation genetics papers:

1. Brand, C. L. & Levine, M. T. Cross-species incompatibility between a DNA satellite and the *Drosophila* Spartan homolog poisons germline genome integrity. *Curr. Biol.* CB 32, 2962-2971.e4 (2022).
2. Barbash, D. A., Awadalla, P. & Tarone, A. M. Functional Divergence Caused by Ancient Positive Selection of a *Drosophila* Hybrid Incompatibility Locus. *PLOS Biol.* 2, e142 (2004).
3. Presgraves, D. C. & Meiklejohn, C. D. Hybrid Sterility, Genetic Conflict and Complex Speciation: Lessons From the *Drosophila simulans* Clade Species. *Front. Genet.* 12, (2021).
4. Fang, S. et al. Incompatibility and Competitive Exclusion of Genomic Segments between Sibling *Drosophila* Species. *PLOS Genet.* 8, e1002795 (2012).
5. Ferree, P. M. & Barbash, D. A. Species-Specific Heterochromatin Prevents Mitotic Chromosome Segregation to Cause Hybrid Lethality in *Drosophila*. *PLOS Biol.* 7, e1000234 (2009).

Reviewer #2 (Remarks to the Author):

[Editor's note: Reviewer 2 co-reviewed this manuscript with one of the reviewers who provided the listed reports. This is part of the Nature Communications initiative to facilitate training in peer review and to provide appropriate recognition for Early Career Researchers who co-review manuscripts.]

Please find our point-by-point response below (Reviewer comments, black; replies, green):

We thank the reviewers for their additional input on our manuscript!

We appreciate the efforts that the authors put into several sets of new experiments and textual modifications. Parts of this manuscript were strengthened as a result.

Thank you.

However, two central and related aspects to this work remain incompletely addressed resulting in significant limitations to the study's main results.

The first is the claim that the timeframe over which the ozone treatment depletes fly pheromones is relevant at the timescale of speciation processes. We had previously encouraged the authors to better quantify the “recovery time” for pheromone profiles to return to pre-exposure levels and to consider how these temporal and population/population genetic processes would play out. We appreciate the additional experiments that were carried out to better define the recovery time for *D. melanogaster* and the other species. The result of these experiments indicate that the flies recover their pheromones quicker than previously suggested - two days instead of five. Though this may be a minor change (but in a direction that suggest a reduced role for ozone impacting pheromones and species boundaries), a significant limitation is that only speculations can be made on this point without a more general quantitative framework to guide our thinking and intuition. The authors have only added verbal arguments in favor of ozone’s importance and have not provided additional quantitative arguments for how the process(es) would play out. As a result, the relevance of the ozone exposure on between-species dynamics is unconvincing.

Here we disagree. Our new data shows that it takes more than 24 hours for flies to recover their degraded pheromones after ozone exposure. As in polluted places ozone has been reported to repeatedly reach peak values on a daily base, any recovery time that is longer than 24 hours will lead to long term effects. We have modified the corresponding text in the result section to “Hence, it takes more than 24 hours for the pheromone levels to recover. As peaks of ozone concentrations often re-occur on a daily base²⁶, flies in polluted areas might not be able to re-establish their pheromone levels in-between.”

The other major limitation in the current draft is that the authors still seem largely unaware of the

relevant literature on speciation genetics directly relevant to questions they raise, including numerous studies that have investigated the genetic reasons that species within the *D. simulans* complex are unable to create new ‘hybrid species’ (as well as the broader literature on the genetics of speciation). In addition to the observation that the species conform to Haldane’s rule, specific genetic incompatibles have been described (see citations below). These genetic incompatibles mean that even if the flies’ pheromone-based recognition systems are altered other genetic barriers would prevent hybrid species from forming. It was also unclear if the authors realize that hybrids with sterile males would only result in gene flow through hybrid:parental species matings which would be unlikely to persist over long timespans given the many more conspecific matings. There are no reasons to suspect that these instances of gene flow would result in new species or the decline of a species in the way the authors suggest. Though the authors provided a considerable amount of new text on the topic of speciation, the hope was that it would have been used to better explain their own thoughts about how their observations/hypothesis in light of these speciation genetic papers and not in such general terms. As the manuscript stands, the conclusions and discussion are naive and too disconnected to the field that they aim to contribute to.

Several relevant speciation genetics papers:

1. Brand, C. L. & Levine, M. T. Cross-species incompatibility between a DNA satellite and the *Drosophila* Spartan homolog poisons germline genome integrity. *Curr. Biol.* CB 32, 2962-2971.e4 (2022).
2. Barbash, D. A., Awadalla, P. & Tarone, A. M. Functional Divergence Caused by Ancient Positive Selection of a *Drosophila* Hybrid Incompatibility Locus. *PLOS Biol.* 2, e142 (2004).
3. Presgraves, D. C. & Meiklejohn, C. D. Hybrid Sterility, Genetic Conflict and Complex Speciation: Lessons From the *Drosophila simulans* Clade Species. *Front. Genet.* 12, (2021).
4. Fang, S. et al. Incompatibility and Competitive Exclusion of Genomic Segments between Sibling *Drosophila* Species. *PLOS Genet.* 8, e1002795 (2012).
5. Ferree, P. M. & Barbash, D. A. Species-Specific Heterochromatin Prevents Mitotic Chromosome Segregation to Cause Hybrid Lethality in *Drosophila*. *PLOS Biol.* 7, e1000234 (2009).

We thank the reviewer for pointing us at those references on genetic incompatibilities between species of the *D. melanogaster* complex. We have now included the references in the manuscript. We agree that reported gene incompatibilities between some species pairs within the *D. melanogaster* complex might impede the chance of ozone-induced hybrid speciation events in those species pairs. We therefore have toned down our conclusions regarding a potential species hybridization within the *D. melanogaster* complex (and have moved parts of the discussion on that to the supplementary discussion, as suggested by the editor). However, more than 100 other sympatric species pairs in *Drosophila* have been reported (Coyne and Orr, Evolution 1989) that can hybridize and where often nothing is known regarding genetic incompatibilities. From our last paper (Jiang *et al.* Nature Communications 2023) we know, that pheromones of many *Drosophila* species become degraded by ozone. We therefore still believe, that the ozone-induced increased frequencies of hybridizations have the potential to pave the road for hybrid speciation between at least some species. We have carefully revised our conclusions regarding a potentially increased chance for hybrid speciation in the main manuscript. We have, however toned down our conclusions throughout the manuscript (all novel parts marked in **green** in the manuscript)

For your better overview, we have listed the occasions where we still mention the potential for hybrid speciation below:

Abstract:

....Such mating advantage potentially could lead to hybrid speciation. Hence, the pollutant-induced breakdown of species boundaries could contribute to both, population decline and novel speciation events during the Anthropocene.

Main:

....This might lead to a continued gene flow between two closely related species and to the formation of stable hybrid populations. Hence, our data suggest that by degrading insect pheromones, Anthropogenic oxidant pollutants via hybrid speciation **potentially** may govern the origin of new species.

Discussion:

... While speciation is an ongoing process throughout evolution, in the Anthropocene increased oxidant pollutants might, hence, **potentially** pave the road for new and unexpected hybrid speciation events.

Supplementary Discussion:

Our manuscript deals with four species of the *Drosophila melanogaster* complex. *D. sechellia*, and *D. mauritiana* most probably have evolved from a large mainland population of a shared ancestor with *D. simulans* through allopatric speciation based on two island colonization events⁶⁴. *D. simulans*, like *D. melanogaster* nowadays is globally distributed and also occurs on Mauritius and the Seychelles, i.e. the islands originally inhabited by *D. mauritiana* and *D. sechellia*. It has been shown that in *Drosophila* flies during speciation usually first prezygotic isolation (i.e. via courtship and mating boundaries) and afterward postzygotic isolation (via hybrid sterility and inviability) are established^{65,66}. *D. simulans*, *D. sechellia*, and *D. mauritiana* belong to the *simulans* species complex and have established prezygotic isolation based on e.g. species-specific pheromonal blends^{18,67-69} and courtship songs⁷⁰⁻⁷². Their post-zygotic isolation, however, is incomplete, as only male hybrids are sterile, while female hybrids are fertile. On both islands, gene flow via hybridization events between *D. simulans* and its close relatives has been reported (with *D. sechellia*³⁶; with *D. mauritiana*⁵²), suggesting that presynaptic isolation between these species is not absolute. Our data reveal, that oxidant pollutants like ozone have the potential to corrupt prezygotic isolation and, hence, make hybridization events more likely. As at least some of the resulting hybrids seem to be competitive regarding mate choice (Fig. 3) and reproduction (Fig. S6), such hybridization events potentially could result in ongoing geneflow between sympatric species. As for these species several genetic incompatibilities have been reported (Brand and Levine Curr. Biol. CB 32, 2962-2971.e4 (2022). Barbash et al. PLOS Biol. 2, e142 (2004). Presgraves and Meiklejohn Front. Genet. 12, (2021), Fang et al. PLOS Genet. 8, e1002795 (2012). Ferree and Barbash PLOS Biol. 7, e1000234 (2009)) it, however, remains open, whether ongoing gene flow in this species complex has the potential to finally result in hybrid speciation.